# Chemogenetics Modulation of Electroacupuncture Analgesia in Mice Spared Nerve Injury-Induced Neuropathic Pain through TRPV1 Signaling Pathway

**DOI:** 10.3390/ijms25031771

**Published:** 2024-02-01

**Authors:** I-Han Hsiao, Chia-Ming Yen, Hsin-Cheng Hsu, Hsien-Yin Liao, Yi-Wen Lin

**Affiliations:** 1School of Medicine, College of Medicine, China Medical University, Taichung 40402, Taiwan; coolfishing2002@gmail.com; 2Department of Neurosurgery, China Medical University Hospital, Taichung 40402, Taiwan; 3Department of Anesthesiology, Taichung Tzu Chi Hospital, Buddhist Tzu Chi Medical Foundation, Taichung 42743, Taiwan; terryyen1974@gmail.com; 4School of Post-Baccalaureate Chinese Medicine, Tzu Chi University, Hualien 97004, Taiwan; 5School of Post-Baccalaureate Chinese Medicine, College of Chinese Medicine, China Medical University, Taichung 40402, Taiwan; hchsu@mail.cmu.edu.tw; 6Department of Traditional Chinese Medicine, China Medical University Hsinchu Hospital, Hsinchu 302, Taiwan; 7Graduate Institute of Acupuncture Science, College of Chinese Medicine, China Medical University, Taichung 40402, Taiwan; 8Chinese Medicine Research Center, China Medical University, Taichung 40402, Taiwan

**Keywords:** neuropathic pain, chemogenetics, TRPV1, pERK, electroacupuncture, SSC

## Abstract

Neuropathic pain, which is initiated by a malfunction of the somatosensory cortex system, elicits inflammation and simultaneously activates glial cells that initiate neuroinflammation. Electroacupuncture (EA) has been shown to have therapeutic effects for neuropathic pain, although with uncertain mechanisms. We suggest that EA can reliably cure neuropathic disease through anti-inflammation and transient receptor potential V1 (TRPV1) signaling pathways from the peripheral to the central nervous system. To explore this, we used EA to treat the mice spared nerve injury (SNI) model and explore the underlying molecular mechanisms through novel chemogenetics techniques. Both mechanical and thermal pain were found in SNI mice at four weeks (mechanical: 3.23 ± 0.29 g; thermal: 4.9 ± 0.14 s). Mechanical hyperalgesia was partially attenuated by 2 Hz EA (mechanical: 4.05 ± 0.19 g), and thermal hyperalgesia was fully reduced (thermal: 6.22 ± 0.26 s) but not with sham EA (mechanical: 3.13 ± 0.23 g; thermal: 4.58 ± 0.37 s), suggesting EA’s specificity. In addition, animals with *Trpv1* deletion showed partial mechanical hyperalgesia and no significant induction of thermal hyperalgesia in neuropathic pain mice (mechanical: 4.43 ± 0.26 g; thermal: 6.24 ± 0.09 s). Moreover, we found increased levels of inflammatory factors such as interleukin-1 beta (IL1-β), IL-3, IL-6, IL-12, IL-17, tumor necrosis factor alpha, and interferon gamma after SNI modeling, which decreased in the EA and *Trpv1*^−/−^ groups rather than the sham group. Western blot and immunofluorescence analysis showed similar tendencies in the dorsal root ganglion, spinal cord dorsal horn, somatosensory cortex (SSC), and anterior cingulate cortex (ACC). In addition, a novel chemogenetics method was used to precisely inhibit SSC to ACC activity, which showed an analgesic effect through the TRPV1 pathway. In summary, our findings indicate a novel mechanism underlying neuropathic pain as a beneficial target for neuropathic pain.

## 1. Introduction

Pain is a global healthcare concern and the most common reason for people seeking medical intervention. Chronic pain is defined as a continued painful sensation lasting for >3 months in the clinic and can deteriorate the quality of life. The International Association for the Study of Pain (IASP) categorized chronic pain as primary and secondary pain [1]. Primary pain includes fibromyalgia and nonspecific lower back pain [2]. Neuropathic pain is chronic pain secondary to an underlying syndrome that can arise from nociceptive input, even without clear damage or dysfunction [1]. Chronic pain is associated with pain signal processing, central sensitization, and neural circuits. Spontaneous hyperalgesia was frequently detected with unsatisfactory therapy for neuropathic pain. Recent studies show that the central nervous system can suffer alterations in central sensitization in mice [3,4]. Pain continues due to central potentiation, a type of neuronal plasticity, and amplifies neural activity after the initiation. Central sensitization also results from inflammation in either the peripheral or central nervous system [5].

Neuropathic pain is demarcated as an uncomfortable sensation initiated by a malfunction of the somatosensory cortex (SSC) induced by central nerve system dysfunction in humans [6]. In addition, strokes in the SSC region can also induce neuropathic pain [7]. Maladaptive neuroinflammation beyond neuronal damage is the dominant cause of chronic pain, even after stroke resolution. Recently, the microglia have been involved in neuropathic pain as neuroinflammation triggers microglial activation [8]. Neuropathic pain is a debilitating condition resulting from central sensitization with increased excitatory inputs from several brain areas, including the SSC, to the anterior cingulate cortex (ACC) in response to sensory-discriminative pain features [4]. For spared nerve injury (SNI) at the sciatica trunk, common peroneal, tibia, or sural nerves are commonly used to develop animal models for drug development [9,10,11,12]

Transient receptor potential V1 (TRPV1) is a Ca^2+^-permeable channel associated with pain sensation and inflammation detection [13]. TRPV1 is triggered by mechanical force, capsaicin, low pH, or temperatures over 43 °C. TRPV1 was indicated to participate in inflammatory, fibromyalgia, and neuropathic pain [14,15] and is highly expressed in small C fibers. TRPV1 is significantly increased after both peripheral and central levels, such as inflammatory, fibromyalgia, and neuropathic pain models. Trpv1 deletion induced desensitization to thermal stimulation and the inflammatory mediators released from glial cells [4]. Moreover, injection of the TRPV1 antagonist reduces thermal pain in mice inflammatory pain. Protease-activated receptor 2 is demonstrated to activate TRPV1 through protein kinase A or C signals [14]. HMGB1 was indicated as a nuclear factor enhancing transcription associated with tissue damage, bacterial infection, and inflammation. HMGB1 was known as inflammatory mediators from necrotic or glial cells activated by interleukins (ILs), tumor necrosis factor α (TNF-α), and interferon-γ (IFN-γ) [16]. HMGB1 can bind to receptors for advanced glycation end-products (RAGE) in the regulation of DNA transcription by nuclear factor kappa-light-chain-enhancer of activated B cells (NF-κB). The S100B was a calcium-wrapping protein that can bind to the RAGE receptor, categorized by the regulation of procedures involved in tissue damage [17].

Acupuncture has been long performed as an alternative cure for pain healing with rare side effects. Recently, electroacupuncture (EA), which provides electric stimulation along with acupuncture for better results and standardization, has been shown to relieve inflammatory, fibromyalgia, and neuropathic pain [18] in several mice models. Moreover, it can alleviate various pain conditions by increasing endogenous opiates [19] and adenosine [20]. Moreover, EA was reported to alleviate fibromyalgia pain by attenuating IL-1β, TNFα, and IFNγ in mice plasma. In addition, we recently showed that EA reliably attenuates either mechanical or thermal hyperalgesia in fibromyalgia mouse models by inhibiting the TRPV1 pathway [4].

In the current research, we hypothesized that EA consistently attenuates neuropathic symptoms via anti-inflammation and TRPV1 signaling pathways. We mimicked neuropathic pain using the SNI mice model and evaluated EA efficacy. We suggested that neuropathic pain is allied with amplified inflammation and neural hyperactivity from the peripheral dorsal root ganglion (DRG), central spinal cord dorsal horn (SCDH), SSC, and ACC. Consistent with our hypothesis, EA treatment reduced SNI-induced neuropathic pain, which also decreased in *Trpv1*^−/−^ mice. TRPV1 and related factors were significantly amplified in the mice DRG, SCDH, SSC, and ACC regions. Conversely, either EA or *Trpv1* gene deletion significantly attenuated this increase. Thus, EA had an anti-nociceptive effect by downregulating TRPV1. Furthermore, we used a novel chemogenetics technique to precisely attenuate hyperactivity in the mice SSC. After SSC inhibition, neuropathic pain and TRPV1 signaling were attenuated both in the SSC and ACC. Thus, we provided an indication that the aforementioned mediators can modify the TRPV1 pathway and indicated novel and prospective healing goals for treating neuropathic pain.

## 2. Results

### 2.1. Electroacupuncture at Mice ST36 Acupoint Ameliorated Mechanical and Thermal Hyperalgesia in a Spared Nerve Injury-Initiated Chronic Pain Model

To understand whether EA can ameliorate neuropathic pain, we used behavioral tests to evaluate treatment efficacy. Before SNI surgery (day 0), all groups presented similar mechanical pain behavior. Three days after SNI, mechanical hyperalgesia was observed in all groups. In addition, all groups became more pain sensitive over time (Figure 1A, SNI: 3.23 ± 0.29 g, n = 9, * *p* < 0.05). We did not provide any interventional treatment until 15 days to initiate chronic pain. After the 15th day, 2 Hz EA partially attenuated mechanical hyperalgesia in the SNI mice model instead of the sham group (Figure 1A, EA: 4.05 ± 0.19 g: sham EA: 3.13 ± 0.23 g, n = 9, * *p* < 0.05). Similarly, partial mechanical nociception was perceived in *Trpv1*^−/−^ mice (Figure 1A, *Trpv1*^−/−^: 4.43 ± 0.26 g, n = 9, * *p* < 0.05). Thermal hyperalgesia showed a similar tendency. All groups except the *Trpv1*^−/−^ mice showed a gradual decrease in latency after two weeks (Figure 1B, SNI: 4.9 ± 0.14 s; n = 9, * *p* < 0.05). After the 15th day, EA significantly reduced thermal hyperalgesia while that in the sham group kept worsening (Figure 1B, EA: 6.22 ± 0.26 s: sham EA: 4.58 ± 0.37 s, n = 9, * *p* < 0.05). In the thermal pain behavior test, there was no hyperalgesia in *Trpv1*^−/−^ mice from the 3rd to 28th day (Figure 1B, *Trpv1*^−/−^, 6.24 ± 0.09 s, n = 9, * *p* < 0.05). Figure 1C shows a schematic illustration of the SNI procedure.

### 2.2. Electroacupuncture at ST36 Restored Elevated SNI-Derived Serum Inflammatory Mediators

We evaluated the concentration of inflammatory mediators in mouse plasma. SNI mice showed increases in inflammatory factors such as IL-1β, IL-3, IL-6, IL-12, IL-17, TNF-α, and IFN-γ than the normal group (Figure 2, * *p* < 0.05, n = 9) and similar to the sham control (Figure 2, * *p* < 0.05, n = 9). The intensities of these mediators were lower in the EA and Trpv1 gene deletion groups (Figure 2, EA group: # *p* < 0.05, n = 9, *Trpv1*^−/−^ group: # *p* < 0.05, n = 9). 

### 2.3. Electroacupuncture Alleviated the Overexpression of Microglial Transmission and TRPV1-Related Kinases in Mice DRG

Western blot was used to investigate microglial and neuronal markers. SNI mice had higher levels of Iba1, a microglial marker (Figure 3A, * *p* < 0.05, n = 6). This effect was diminished in the EA and *Trpv1^−/−^* groups rather than the sham group (Figure 3A, # *p* < 0.05, n = 6). We next measured HMGB1 and S100B in mice DRG. Similar to Iba1, their expressions were augmented in the SNI group (Figure 3A, * *p* < 0.05, n = 6) than the EA and *Trpv1*^−/−^ groups (Figure 3A, # *p* < 0.05, n = 6). The overexpression of HMGB1 or S100B did not alter in sham mice (Figure 3A, * *p* < 0.05, n = 6). In addition, the SNI group had higher TRPV1 levels (Figure 3A, * *p* < 0.05, n = 6), but 2 Hz EA meaningfully abridged TRPV1 levels (Figure 3A, # *p* < 0.05, n = 6). TRPV1 was almost disappeared in *Trpv1^−/−^* mice. The SNI group had increased levels of pPI3K, pAkt, and pmTOR (Figure 3B, * *p* < 0.05, n = 6) than EA or *Trpv1^−/−^* mice (Figure 3B, # *p* < 0.05, n = 6). We also confirmed the involvement of the MAPK pathway by observing higher levels of pERK, pp38, and pJNK in SNI mice (Figure 3B yellow column and 3C black and light gray columns, * *p* < 0.05, n = 6). Their levels were significantly attenuated in the EA or *Trpv1^−/−^* groups compared to the SNI group (Figure 3B yellow column and 3C black and light gray columns, # *p* < 0.05, n = 6). Furthermore, we observed higher expressions of the transcription factors pNFκB and pCREB in the DRG in the SNI group than in the normal (Figure 3C deep gray and yellow columns, * *p* < 0.05, n = 6), EA, and *Trpv1^−/−^* groups (Figure 3C deep gray and yellow columns, # *p* < 0.05, n = 6). Immunofluorescence staining indicated TRPV1 appearance in the mice DRG. TRPV1 levels significantly increased in the SNI model, an increase attenuated by EA treatment and *Trpv1^−/−^* mice (Figure 3D, n = 3). Next, immunostaining the DRG with Iba1, an antibody showed higher Iba1 expression in the DRG of SNI mice than attenuated in EA or *Trpv1^−/−^* mice (Figure 3E, n = 3). When these two staining images were merged, the SNI and sham groups showed obvious fluorescence compared to the control, 2Hz EA, and *Trpv1^−/−^* groups (Figure 3F, n = 3).

### 2.4. Electroacupuncture Alleviated Spared Nerve Injury-Induced Microglial and TRPV1 Increased Appearance in the Spinal Cord Dorsal Horn 

The spinal cord is the main region that conveys pain sensation via the dorsal horn, which is the starting central sensitization. We observed increased expressions of Iba1, HMGB1, S100B, and TRPV1 in the SCDH after SNI (Figure 4A, * *p* < 0.05, n = 6), which decreased after 2 Hz EA (Figure 4A, # *p* < 0.05, n = 6). In contrast, sham EA did not change the levels of these proteins. In *Trpv1^−/−^* mice (Figure 4A), the levels of Iba1, HMGB1, and S100B (Figure 4A, # *p* < 0.05, n = 6) were significantly lower. EA treatment and *Trpv1* deletion alleviated the intensification in the pPI3K-pAkt-pmTOR pathway observed in the SNI mice (Figure 4B, * *p* < 0.05, n = 6). In addition, SNI increased pERK, pp38, and pJNK expressions (Figure 4B, * *p* < 0.05, n = 6), a consequence reversed by EA or Trpv1 deletion (Figure 4B,C, # *p* < 0.05, n = 6). The same tendency was also perceived for pNFκB and pCREB (Figure 4C). Next, immunostaining in the SCDH showed an increase in TRPV1 expression in the SNI group (Figure 4D, n = 3), in which 2 Hz EA and *Trpv1^−/−^* decreased, while the sham group showed similar fluorescence intensity as the SNI group (Figure 4D, n = 3). As shown in Figure 4E (n = 3), we observed a similar pattern for Iba1. We further detected amplified dual staining in the SNI mice SCDH, suggesting co-localization of TRPV1 and Iba1 (Figure 4F, n = 3). EA or Trpv1 deletion removed the aforementioned indications (Figure 4F, n = 3).

### 2.5. Electroacupuncture Improved Microglial Hyperfunction and TRPV1 Activation in the Somatosensory Cortex after SNI Induction

To determine the microglial or neuronal mechanisms by which TRPV1 modulates SNI-induced neuropathic pain, we detected Iba1 and TRPV1 levels in the mouse SSC. As shown in Figure 5A, the SNI mice had increased Iba1, HMGB1, S100B, and TRPV1 (Figure 5A, * *p* < 0.05, n = 6), whereas 2 Hz EA downregulated the levels of these four proteins (Figure 5A, # *p* < 0.05, n = 6). Similar results were observed in the *Trpv1^−/−^* group instead of the sham group. Then, we explored related downstream factors such as pERK, pp38, and pJNK expressions in the cytoplasm and the transcription factors pNFκB and pCREB in the nucleus. All were increased in the SNI group and decreased after 2 Hz EA and Trpv1 deletion. Immunostaining showed similar expression levels for Iba1 (Figure 5D, n = 3) and TRPV1 (Figure 5E, n = 3) in the mice SSC area and increased dual-stained signals in the SNI mice, proposing TRPV1 and Iba1 co-localization (Figure 5F, n = 3), an effect attenuated by EA or Trpv1 deletion, rather than sham EA.

### 2.6. Electroacupuncture Reversed Microglial Activity, TRPV1, and Associated Molecules Changes in the ACC after SNI

ACC is the terminal region in the pain transmission pathway. As shown in Figure 6A, EA and Trpv1 deletion significantly alleviated the increased Iba1, HMGB1, S100B, and TRPV1 levels after SNI in this region (Figure 6A, * *p* < 0.05, n = 6). Downstream proteins in mice ACC, such as pPI3K-pAkt-pmTOR, pERK, pJNK, pp38, pNFκB, and pCREB, were all augmented in the SNI group, but their levels were restored after 2 Hz EA and Trpv1 loss (Figure 6B,C, # *p* < 0.05, n = 6). Immunostaining for Iba1 or TRPV1 in the ACC showed that the SNI group showed higher fluorescence intensity than the control, an effect attenuated by 2 Hz EA and *Trpv1* deletion (Figure 6D,E, n = 3). The sham group presented similar intensity as the SNI group. The merged image showed amplified dual-stained signals in the SNI mice and sham group compared to the control group, which was diminished by EA or Trpv1 deletion (Figure 6F).

### 2.7. Chemogenetic Inhibition of SSC Improved Neuropathic Pain in the Mouse SNI Model

Figure 7A shows a significant SNI-induced mechanical hyperalgesia from days 0 to 28 after induction (Figure 7A, black circle, n = 9). In addition, chemogenetic inhibition of the SSC significantly attenuated mechanical hyperalgesia (Figure 7A, red circle, n = 9). With respect to thermal pain, mice under SNI induction showed significant thermal hyperalgesia compared to basal conditions (Figure 7B, black circle, n = 9). Furthermore, after CNO injection, thermal hyperalgesia was relieved in mice with SSC inhibition (Figure 7B, red circle, n = 9). Moreover, our data indicated that microglial and TRPV1-associated molecules were not altered in DRG or SCDH of SNI mice and SNI mice subjected to chemogenetic manipulation (Figure 7C,D, n = 6). In the SSC mice, Iba1, HMGB1, and S100B levels were not altered after CNO injection. Interestingly, TRPV1 and related molecules were attenuated after chemogenetic manipulation (Figure 7E, * *p* < 0.05, n = 6). A comparable tendency was observed in the mice ACC (Figure 7F, * *p* < 0.05, n = 6).

## 3. Discussion

When acute nerve damage progresses to a chronic stage, neuropathic pain develops, which gradually deteriorates from nociception to co-morbid emotional and cognitive discomfort [7,21]. Central nervous sensitization, induced by peripheral nerve injury, contributes to chronic neuropathic pain, indicating a frequency-dependent increase in hyperexcitability of spinal and brain neurons due to peripheral nerves or tissue injury [21]. This sensitization persists via continuous input of a peripheral noxious stimulation signal. Recently, neuroinflammation and associated molecules have been studied to understand neuropathic pain. When the peripheral nerve is damaged, it produces inflammatory stimuli that end up causing neuronal cell death due to oxidative stress. Microglia can secrete MCP-1, IL-1β, IL-6, IL-8, IL-17A, IL-17F, IFN-γ, and TNF-α that interact with each other and with immune T cells [22]. Kiguchi et al. reported that IL-1β mRNA in macrophages and Schwann cells was increased in the damaged sciatic nerve by partial sciatic nerve ligation (PSL). Neuropathic pain can be reduced by injection of IL-1β antibody [23]. A recent paper indicated that injection of IL-1β and TNF-α at the sciatic nerve significantly induced neuropathic pain. Kim and Taylor determined that IL-17 can moderate microglial activation after nerve injury-induced neuropathic pain [24]. This, combined with immune responses, induces unusual synaptic trimming, demyelination, axonal disintegration, regulation of BBB permeability, and immune cell recruitment [5]. These mechanisms ultimately result in the suppression of neurogenesis and neuron apoptosis, even neuronal death.

The IASP recommends tricyclic antidepressants, gabapentin, pregabalin, duloxetine, and venlafaxine as effective drugs for neuropathic pain [25]. These oral drugs influence whole-body synaptic conduction and, hence, result in systemic side effects, including drowsiness, dizziness, and mouth dryness. A Topical patch with capsaicin is also recommended. For clinical usage, topical capsaicin (8%) patches, applied for 60 min, can inhibit the symptoms for 3 months [26]. The mechanism behind capsaicin-based relief from neuropathic pain is associated with the TRPV1 receptor, found on neuronal cell membranes, as capsaicin is a TRPV1 agonist. Theoretically, a high concentration of capsaicin binds to TRPV1 and activates it, opening calcium ion channels and leading to a massive influx of calcium ions [27]. Then, the neurons degenerate and thus relieve the pain. However, after three months, the neurons regenerate, and the pain comes back. However, said treatment methods have many side effects, causing frequent relapses. In comparison, acupuncture is ideal for treating neuropathic pain as it has fewer side effects [28].

In neuropathic pain, neuron and microglia interactions are critical for central sensitization and further develop into chronic pain. However, the association among neurons, microglial, and chronic neuropathic pain remains uncharacterized. Yi et al. reported that microglial inhibition attenuated spinal nerve damage convinced microglial activation and chronic pain. Microglial inhibition also reduced the transcription factor interferon regulatory factor 8 (IRF8) and interleukin 1 beta (IL-1β). They also indicated that reduced microglial function alleviated synaptic transmission following SNT through in vivo spinal cord recording [29]. In addition, chemogenetic blockade of the connection of the locus coeruleus neuron to the basolateral amygdala decreased chronic pain and anxiety and enhanced fear learning, whereas its activation dramatically initiated anxiety-aversive learning and memory index [30]. Furthermore, CNO administration to mice that had a Gi-DREADD in sensory neurons showed increased latency of paw withdrawal for hot stimuli and indicated an analgesic effect [31]. Hence, we validated that chemogenetic inhibition of the SSC significantly reduced the overexpression of elements in the TRPV1 pathway and downregulated these phenomena in downstream ACC areas. The main limitation of the current research indicated that microglial and TRPV1 signaling pathways were merely perceived in the SNI-induced neuropathic pain mice model. Clinical trials are necessary to confirm our present data. We only determine the TRPV1 receptor at the neuronal level. Further study may focus on other receptors in this model, such as TLR2, TLR4, or RAGE receptors (Figure 8). Our data offer evidence of how TRPV1 and associated factors contribute to peripheral and central levels of this neuropathic pain model. 

## 4. Materials and Methods

### 4.1. Animals

All mice conducted here were agreed upon by the Institute of Animal Care and Use Committee of China Medical University (Permit no. CMUIACUC-2022-408), Taiwan, next the conductor for the use of mice (National Academy Press). Our investigation was designed to include n = 9 (n = 6 for Western blot and n = 3 for immunofluorescence). This study required 45 male C57BL/6 mice, including wild type and *Trpv1^−^*^/−^ mice (Jackson Lab, Bar Harbor, ME, USA) 8–12 weeks’ age with body weightiness of 20–25 g. Mice had been kept at a 12 h dark and light sequence with food and water ad libitum. A model size of 9 mice in each group was essential for an α value of 0.05 with a power of 80%. The laboratory worker blindly followed treatment allocation during analysis and experiment. We divided the mice into 5 groups: Normal group (Group 1: Normal): Mice suffered a skin incision and muscle dissection without nerve incision; Spared nerve incision group (Group 2: SNI): Mice underwent spared nerve incision; EA group (Group 3: SNI + EA): SNI mice with 2 Hz EA treatment; Sham EA group (Group 4: Sham EA): SNI mice with sham treatment; *Trpv1*^−/−^ group (Group 5: *Trpv1*^−/−^): *Trpv1* knockout mice with spared nerve incision.

### 4.2. Neuropathic Pain Model

We used an SNI model to mimic neuropathic pain in animals (15). Mice were anesthetized with a mixture of Zoletil 50 (50 mg/mL) and xylazine (10 mg/mL). After shaving the left side of the thigh and using the femur as a landmark, a linear incision was made along the femoral bone to perform a blunt dissection through the biceps femoris muscle to disclose the three branches of the sciatic nerve. The sural nerve, the so-called spared nerve, is the thinnest among these branches and needs to be carefully protected. The tibia and common peroneal nerves were split over 2 mm and were removed to induce neural damage and inflammation. In the normal group, only femoris muscle dissection was performed, and exposure of the sciatic nerve and its branches was without dissection.

### 4.3. Electroacupuncture

The 2 Hz EA group received acupuncture on the ST36 of the bilateral leg and electrical stimulation with a Trio 300 stimulator (Ito, Tokyo, Japan). The parameters were an intensity of 1 mA for 20 min at 2 Hz with a rhythm length of 100 μs, while the sham group only received needle insertion on ST36 without electrical stimulation. ST36 acupoint was selected since it was frequently conducted in traditional Chinese medicine for the handling of numerous categories of pain. Acupuncture needles (1 inch, 36G; YU KUANG, New Taipei city, Taiwan) were consensually implanted at a depth of 3–4 mm into the murine ST36 acupoint. Mice ST36 acupoint is positioned roughly 3–4 mm lower and 1–2 mm adjacent to the center of the knee [4]. The EA stimulation lasted for two weeks between the third week after SNI surgery and until the end of the fourth week.

### 4.4. Nociceptive Behavioral Tests

The hyperalgesic activities were evaluated twice per week for four weeks after SNI initiation. The first test was three days after SNI surgery. Animals were transferred to the behavior examination area at room temperature and adjusted to the surroundings for ≥30 min. The animals were stimulated after they calmed down and were not sleeping or grooming. In the von Frey filament assessment, mice had been positioned at a steel net (75 × 25 × 45 cm) sheltered with a glass cage (10 × 6 × 11 cm). We used an automated von Frey tester (IITC Life Science Inc., Woodland Hills, CA, USA) to stimulate each mouse three times with the fiber at the hind paw (lesion side) and recorded the force after paw withdrawal. After the mechanical tests, we allowed mice to calm down for ≥30 min before the Hargreaves test, for which we used an IITC algesimeter (IITC Life Sciences, SERIES8, Model 390G, CA, USA) to examine thermal hyperalgesia by counting the latency to heat challenge. The mice stood on a pane sheet in a cage under which there was a heater; the emphasis of the prediction bulb was expected exactly at the left hind paw. In this test, the thermal hyperalgesic latency was judged by the time of paw removal under heat stimulation [4].

### 4.5. Western Blot Analysis 

Mice were euthanized with isoflurane and then underwent cervical dislocation. DRG, SCDH, SSC, and ACC proteins had been used to pluck proteins. Proteins were first kept at 4 °C and then stored at −80 °C. For protein mining, all samples were standardized in radioimmunoprecipitation (RIPA) lysis solution having 50 mM Tris-HCl, 1% NP-40, 250 mM NaCl, 50 mM NaF, 5 mM EDTA, 1 mM Na_3_VO_4_, 0.02% NaN_3_, and 1× protease inhibitor cocktail (AMRESCO). The mined proteins were run with 8% SDS-Tris glycine gel electrophoresis and relocated to a polyvinylidene difluoride (PVDF) membrane. The membrane was incubated in bovine serum albumin (BSA) in TBS-T buffer (10 mM Tris pH 7.5, 100 mM NaCl, 0.1% Tween 20), followed by incubation with 1st antibody in TBS-T with 1% BSA over 2 h in a refrigerator. Peroxidase-conjugated antibody (1:5000) was used as a suitable secondary antibody. Significant bands were imaged using an enhanced chemiluminescent test (PIERCE) via LAS-3000 Fujifilm (Fuji Photo Film Co., Ltd., Tokyo, Japan). The concentration of specific bands in the image was measured with NIH ImageJ 1.54h (Bethesda, MD, USA). A-tubulin or β-actin were used as an interior control [4]. 

### 4.6. Immunofluorescence

After all behavioral tests, mice were euthanized for sample collection by isoflurane inhalants and perfused with 4% paraformaldehyde. The DRG, SC, SSC, and ACC regions were instantly removed and post-fixed with paraformaldehyde in a refrigerator for a few days. Then, such samples were transferred to a 30% sucrose solution for cold protection at 4 °C in a refrigerator overnight. These fixed tissues were placed in an optimal cutting temperature (OCT) compound, fast-frozen with liquid nitrogen, and stored at −80 °C. Stored sections were next cut with 20 μm thickness and rapidly engaged on glass slides. The slides were then Incubated with paraformaldehyde and nurtured with a liquid consisting of 2% BSA, 0.2% Triton X-100, and 0.03% sodium azide over 1 h at room temperature. After blocking, the slices were then nurtured with 1st antibody (1:200, Alomone, Israel) in a 1% BSA solution at 4 °C in a refrigerator. Next, slides were further nurtured with 2nd antibody (1:500) for 2 h at room temperature before fixing with coverslips for imaging under a fluorescent microscope (Olympus, BX-51, Tokyo, Japan).

### 4.7. Chemogenetic Operation

All rodents were anesthetized with isoflurane, and we next fixed their heads in a stereotaxic device. A 23-gauge, 2 mm stainless cannula was implanted into SSC, 0.5 mm posterior and 1.5 mm lateral of bregma at 175 μm below the cortical superficial and fixed to the skull with dental glue. The injection cannula was implanted and linked with a Hamilton needle via a PE duct to inoculate 0.3 μL of viral liquid for more than 3 min through the pump (KD Scientific, Holliston, MA, USA). After injection, the shot cannula was preserved at SSC for an extra 2 min to permit the liquid to diffuse. Next, 0.3 μL of hM4D DREADD (designer receptors exclusively activated by designer drugs: AAV8-hSyn-hM4D(Gi)-mCherry; Addgene Plasmid #50477, Watertown, MA, USA) were injected into the SSC over two weeks. Clozapine N-oxide (CNO; Sigma C0832, St. Louis, MO, USA) was injected to stimulate the DREADD. CNO was thawed in 5% dimethyl sulfoxide (DMSO; Sigma D2650) and diluted with normal saline before intraperitoneal injection of 1 mg/kg at day 15.

### 4.8. Statistical Analyses

Arithmetical analysis was achieved via SPSS 21 software. All value results were offered as mean ± standard error (SEM). Shapiro–Wilk examination was accomplished to check the normality of the obtained results (*p* = 0.873, 0.857, 0.511, 0.961, and 0.814 of five groups in von Frey test; *p* = 0.161, 0.560, 0.153, 0.618, and 0.653 of five groups in Hargraves test). Statistical consequence among every group was verified by a one-way ANOVA test, surveyed by post hoc Tukey’s test. Values of *p* < 0.05 were reflected as statistically noteworthy.

## 5. Conclusions

In the present study, the behavior test showed partially increased force in the electronic von Frey filament (mechanical pain) and entire extended latency in the Hargreaves examination (thermal pain) with 2 Hz EA, showing that it can reverse neuropathic pain similarly to a *Trpv1* deletion. Accordingly, we hypothesized that the TRPV1 pathway is crucial in chronic neuropathic pain. We detected blood inflammatory factors and observed that EA can inhibit neuroinflammatory factors, i.e., IL-1β, IL-3, IL-6, IL-12, IL-17, IFN-γ, and TNF-α. Then, we used Western blotting and immunofluorescence to show that 2Hz EA can attenuate microglia and TRPV1 proteins in DRG, SCDH, SSC, and ACC, whereas *Trpv1* deletion mice had the same presentation. In summary, we developed an SNI model to mimic neuropathic pain and indicated that EA can attenuate neuropathic pain (Figure 8).

## Figures and Tables

**Figure 1 ijms-25-01771-f001:**
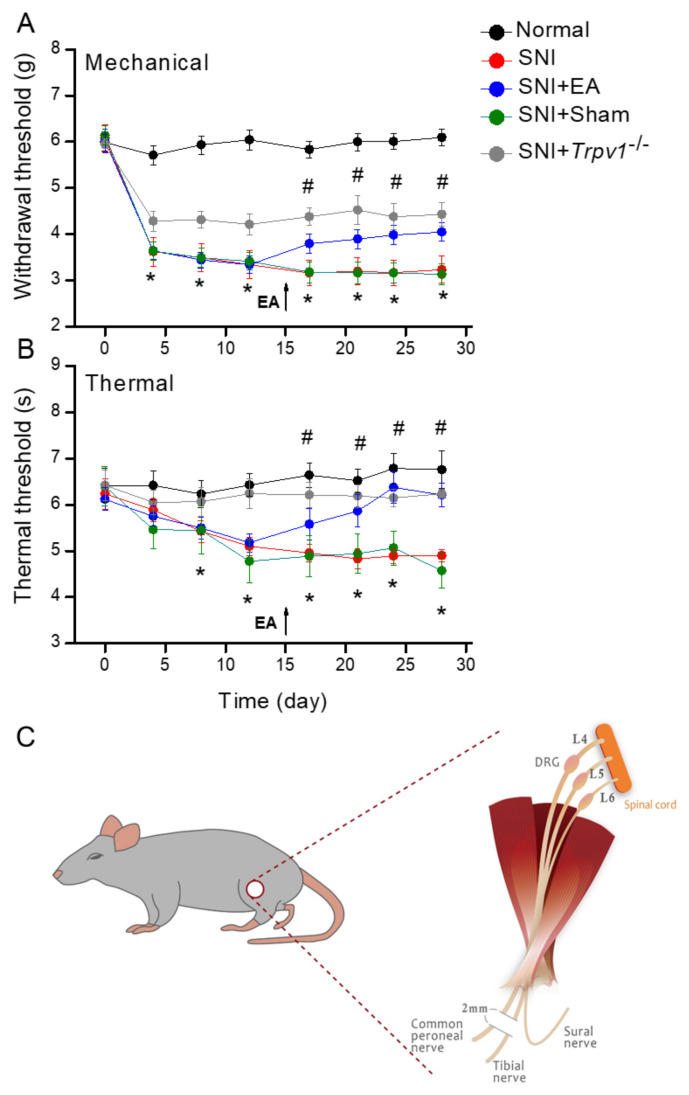
Nociceptor behavior trends. Black: Normal group, red: SNI group, blue: SNI with 2Hz electroacupuncture (SNI + EA) group, green: SNI with Sham group (SNI + Sham EA), and orange: Transient receptor vanilloid member 1 deletion (SNI + *Trpv1*^−/−^) group. From day 0, each node represents post-surgery days 4, 8, 12, 17, 21, 24, and 28. * *p* < 0.05 when compared with the normal group. # *p* < 0.05 when compared with the SNI group. (**A**) Mechanical hyperalgesia (von Frey test). (**B**) Thermal hyperalgesia (Hargreaves test). (**C**) Schematic showing the SNI procedure. First, dissecting the muscle and exposing the sciatica nerve with its three branches: common peroneal nerve, tibial nerve, and sural nerve. Carefully protect and preserve the sural nerve, i.e., spared nerve and cutting another two branches 2 mm at the distal end. Arrow means the initiation of EA.

**Figure 2 ijms-25-01771-f002:**
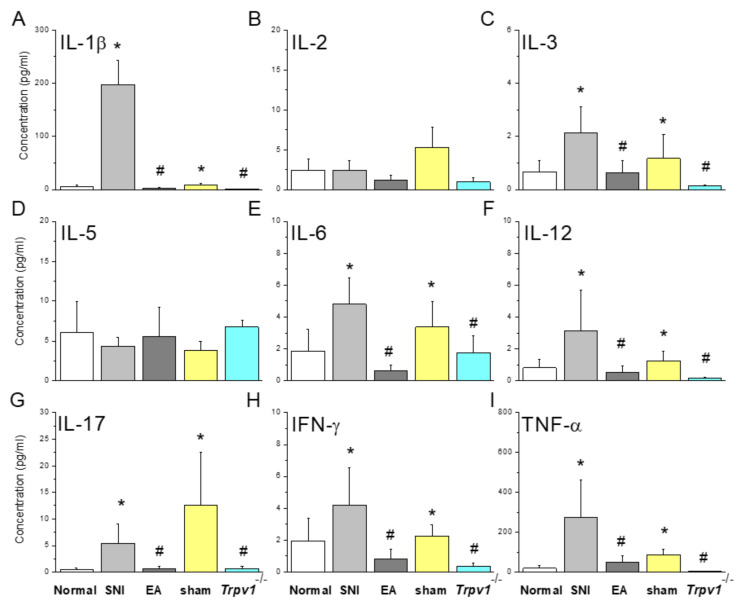
Inflammatory mediators in mice plasma detected by multiplex ELISA. Inflammatory mediators including (**A**) IL-1β, (**B**) IL-2, (**C**) IL-3, (**D**) IL-5, (**E**) IL-6, (**F**) IL-12, (**G**) IL-17, (**H**) IFN-γ, and (**I**) TNF-α. * Indicates statistical significance when compared with the normal group. # indicates statistical significance when compared with the SNI group. IL: interleukin, IFN: interferon, TNF: tumor necrosis factor. n = 9 in all groups.

**Figure 3 ijms-25-01771-f003:**
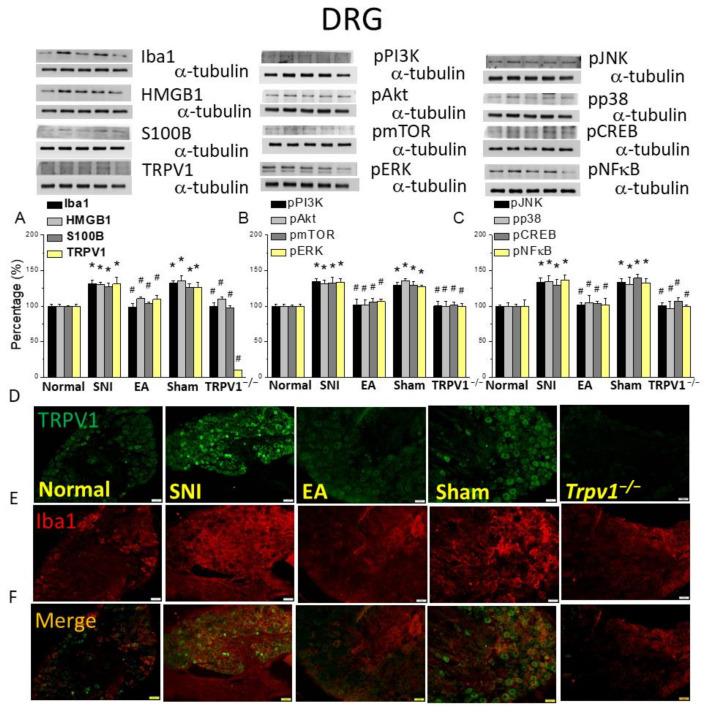
Levels of signaling molecules involved in TRPV1 signaling in the mice DRG. Western blot showing protein expression in the normal, SNI, EA, sham, and *Trpv1^−/−^* groups. Protein levels of (**A**) Iba1, HMGB1, S100B, and TRPV1; (**B**) pPI3K, pAkt, pmTOR, and pERK; and (**C**) pJNK, pp38, pCREB, and pNFkB. * *p* < 0.05 compared with the normal group. # *p* < 0.05 compared with the SNI group. n = 6 in all groups. Immunofluorescence staining of Iba1, TRPV1, and double staining in the mice DRG. (**D**) Iba1, (**E**) TRPV1, and (**F**) Iba1/TRPV1 double staining, immuno-positive (green, red, or yellow) signals in the mouse DRG. Scale bar: 100 μm. n = 3 in all groups.

**Figure 4 ijms-25-01771-f004:**
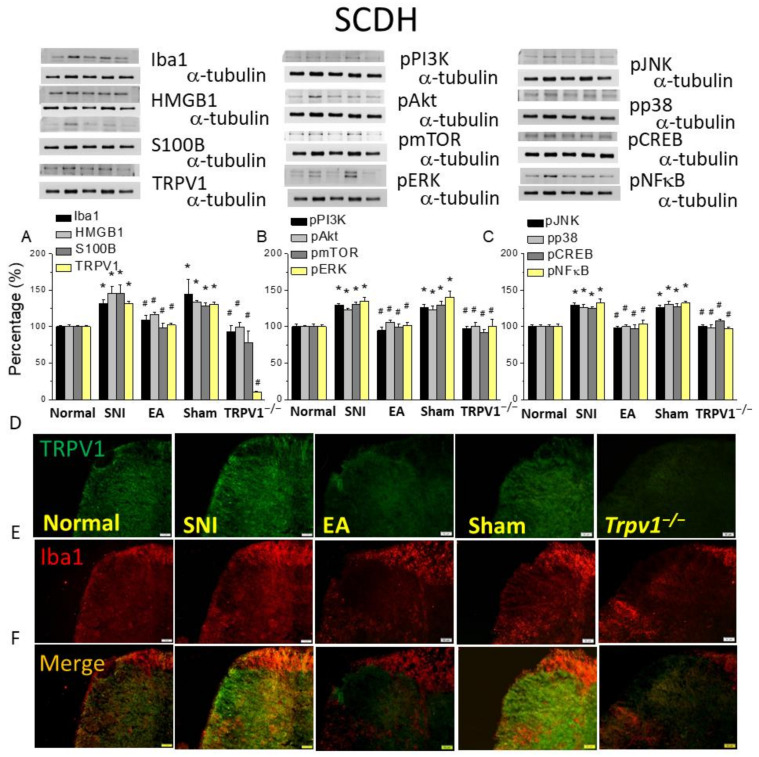
Levels of signaling molecules involved in TRPV1 signaling in the mice SCDH. Western blot protein expression from the normal, SNI, 2Hz EA, sham, and *Trpv1^−/−^* groups. Protein levels of (**A**) Iba1, HMGB1, S100B, and TRPV1; (**B**) pPI3K, pAkt, pmTOR, and pERK; and (**C**) pJNK, pp38, pCREB, and pNFkB. * *p* <0.05 compared with the normal group. # *p*< 0.05 compared with the SNI group. n = 6 in all groups. Immunofluorescence staining of Iba1, TRPV1, and double staining in the mice SCDH. (**D**) Iba1, (**E**) TRPV1, and (**F**) Iba1/TRPV1 double staining, immuno-positive (green, red, or yellow) signals in the mice SCDH. Scale bar: 100 μm. n = 3 in all groups.

**Figure 5 ijms-25-01771-f005:**
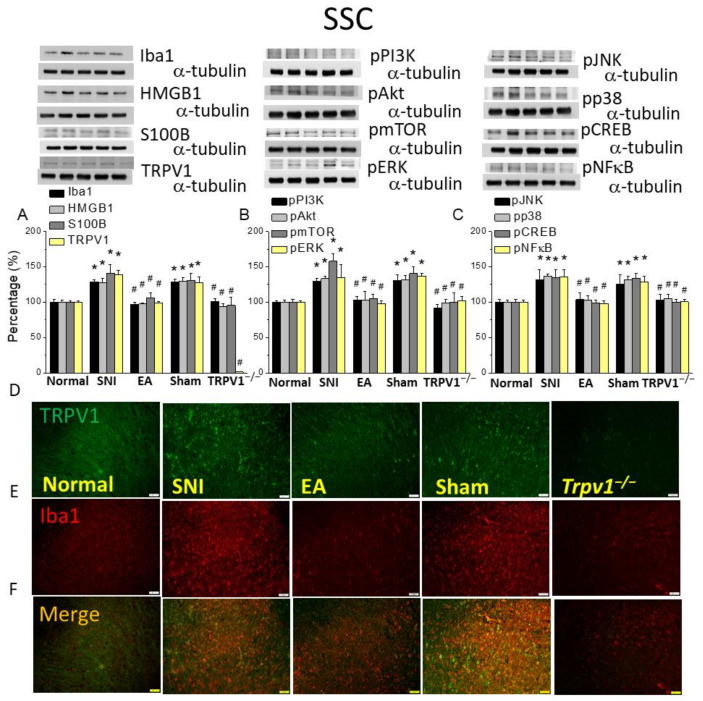
Levels of signaling molecules involved in TRPV1 signaling in the somatosensory cortex (SSC) of the mice. Western blot showing protein expression from the normal, SNI, 2Hz EA, sham, and *Trpv1^−/−^* groups. Protein levels of (**A**) Iba1, HMGB1, S100B, and TRPV1; (**B**) pPI3K, pAkt, pmTOR, and pERK; and (**C**) pJNK, pp38, pCREB, and pNFkB. * *p* < 0.05 compared with the normal group. # *p* < 0.05 compared with the SNI group. n = 6 in all groups. Immunofluorescence staining of Iba1, TRPV1, and double staining in the mouse SCC. (**D**) Iba1, (**E**) TRPV1, and (**F**) Iba1/TRPV1 double staining, immuno-positive (green, red, or yellow) signals in the mice DRG. Scale bar: 100 μm. n = 3 in all groups.

**Figure 6 ijms-25-01771-f006:**
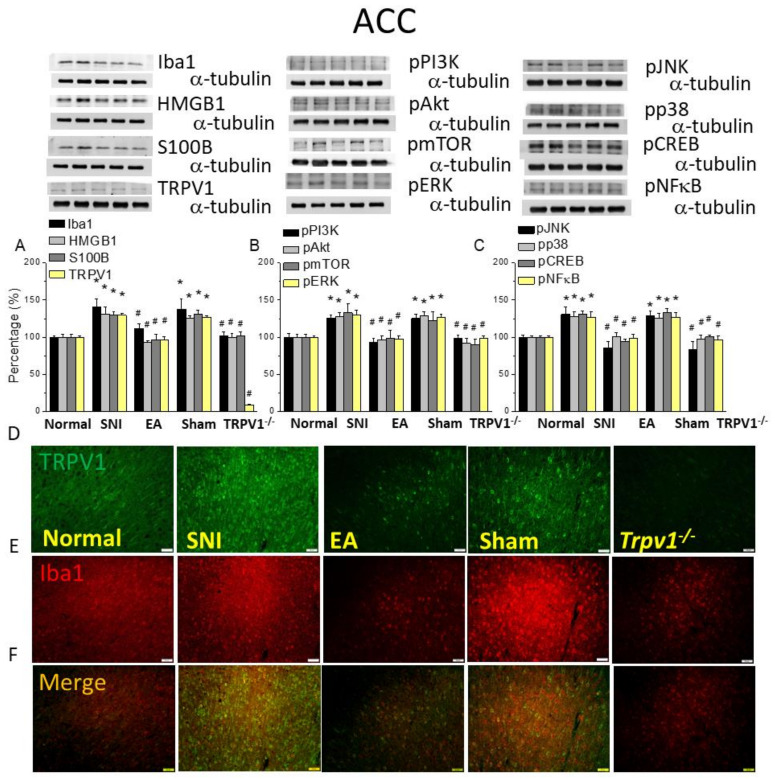
Levels of signaling molecules involved in TRPV1 signaling in the mice ACC. Western blot showing protein expression from the normal, SNI, 2Hz EA, sham, and *Trpv1^−/−^* groups. Protein levels of (**A**) Iba1, HMGB1, S100B, and TRPV1; (**B**) pPI3K, pAkt, pmTOR, and pERK; and (**C**) pJNK, pp38, pCREB, and pNFkB. * *p* < 0.05 compared with the normal group. # *p* < 0.05 compared with the SNI group. n = 6 in all groups. Immunofluorescence staining of Iba1, TRPV1, and double staining in the mice DRG. (**D**) Iba1, (**E**) TRPV1, and (**F**) Iba1/TRPV1 double staining, immuno-positive (green, red, or yellow) signals in the mice DRG. Scale bar: 100 μm. N = 3 in all groups.

**Figure 7 ijms-25-01771-f007:**
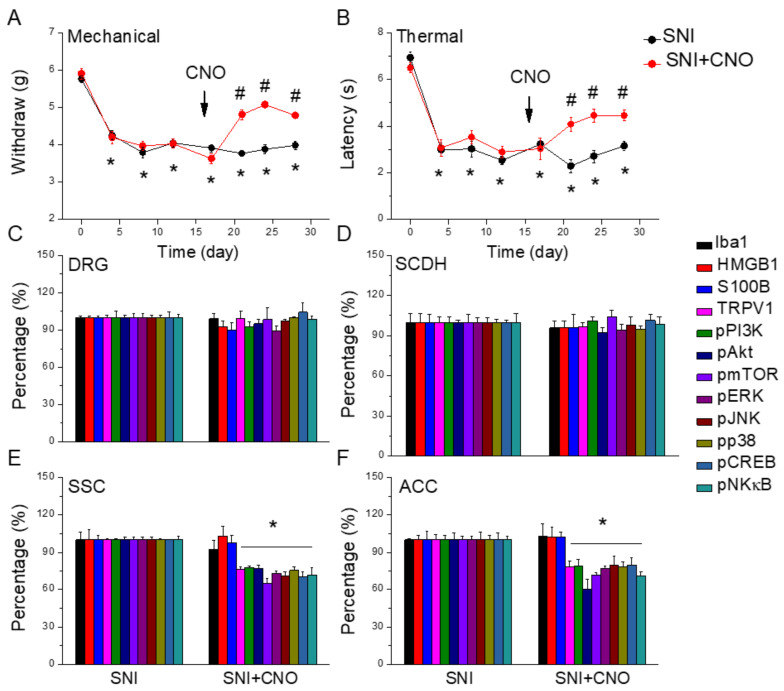
Nociceptor behavior of SNI and SNI animals treated with the chemogenetic technique. Black: SNI group, red: SNI treated with chemogenetics. From day 0, each circle represents post-surgery days 4, 8, 12, 17, 21, 24, and 28. * *p* < 0.05 compared with the normal group. # *p* < 0.05 compared with the SNI group. (**A**) Mechanical hyperalgesia (von Frey test). (**B**) Thermal hyperalgesia (Hargreaves test). Protein levels of Iba1, HMGB1, S100B, TRPV1, pPI3K, pAkt, pmTOR, pERK, pJNK, pp38, pCREB, and pNFkB were measured in mice (**C**) DRG, (**D**) SCDH, (**E**) SSC, and (**F**) ACC. Arrow means the injection of CNO.

**Figure 8 ijms-25-01771-f008:**
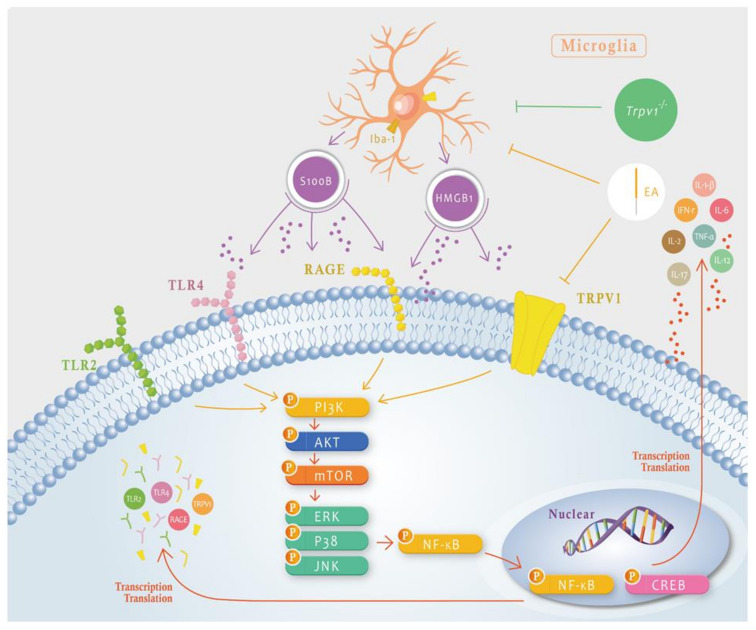
Schematic illustration of the neuron–microglia interaction mechanisms underlying the 2 Hz EA-mediated analgesic effect on SNI-induced neuropathic pain. The summary diagram shows the importance of mechanisms involving microglia and TRPV1 in neuropathic pain. EA can directly inhibit microglial activity, i.e., Iba1 presentation that will result in HMGB1 and S100B release or directly inhibit TRPV1 on the membrane. *Trpv1*^−/−^ mice have the same phenotype as mice treated with EA. P means phosphrylation of kinase.

## Data Availability

The datasets supporting the conclusions of this article are included within the article.

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
