# Peer review of "Chemogenetics Modulation of Electroacupuncture Analgesia in Mice Spared Nerve Injury-Induced Neuropathic Pain through TRPV1 Signaling Pathway"

_ijms, 2024, doi:10.3390/ijms25031771_

Round 1
Reviewer 1 Report
Comments and Suggestions for Authors
Dear Editor,
The manuscript titled "Chemogenetics Modulation of Electroacupuncture Analgesia in Mice Spared Nerve Injury-Induced Neuropathic Pain through TRPV1 Signaling Pathway." presents some interesting results regarding the potential role of TRPV1 in EA-mediated analgesia, I regret to inform you that I cannot recommend its acceptance for publication in its current form.
Major concerns:
Unclear and shifting hypothesis: The stated hypothesis lacks focus and appears to change throughout the manuscript, making it difficult to assess the experimental design and results against a clear objective. A revised hypothesis is crucial for providing direction and coherence to the work.
Questionable novelty: The study appears to be a compilation of three distinct experiments – the effects of EA on SNI pain, changes in TRPV1 signaling in SNI mice, and chemogenetic modulation of SNI – none of which are particularly novel on their own. Combining these elements does not automatically elevate the novelty of the overall work.
Inconsistent data and interpretation: The statement in the abstract and Result section regarding the lack of neuropathic pain induction in Trpv1-/- mice contradicts the data presented in Figure 1A. This discrepancy raises serious concerns about the accuracy and reliability of the findings.
Lack of connection between results and claims: The conclusions drawn from the data do not adequately support the overstated claims in the title and abstract. The relationship between EA, TRPV1, and SNI pain relief needs to be more convincingly established through robust interpretation and analysis.
Recommendations:
The authors should significantly revise the manuscript to address the concerns outlined above. This includes clearly defining a focused hypothesis, demonstrating novelty in the combined approach, resolving the discrepancies in data and interpretation, and strengthening the connection between results and claims.
A thorough reorganization of the manuscript may be necessary to improve the flow and coherence of the presented information.
Despite these limitations, I believe the study holds potential for further exploration. I encourage the authors to address the critical issues identified and resubmit a revised manuscript for consideration.
Thank you for your time and consideration.
Sincerely,
Song Cai
Comments on the Quality of English Languageno comment
Reviewer 2 Report
Comments and Suggestions for Authors
This manuscript discusses the use of electroacupuncture (EA) for treating neuropathic pain in a mice model with a spared nerve injury (SNI). The study found that 2 Hz EA reduced hyperalgesia in the mice, while sham EA did not, indicating the specificity of EA. Mice with a deletion of the transient receptor potential V1 (TRPV1) did not show significant induction of neuropathic pain, further supporting the involvement of TRPV1 in pain. The study also observed increased levels of inflammatory factors in the SNI model, which decreased in the EA and TRPV1-/- groups. Additionally, a novel chemogenetics method was used to inhibit activity in specific brain regions, demonstrating an analgesic effect through the TRPV1 pathway. Overall, the findings suggest a novel mechanism for neuropathic pain and potential targets for its treatment.
· Please add why male animals were used and not both sexes. The ARRIVE guidelines recommend using both sexes in order to find if a sex-related difference exists.
· Please define and add why hyperalgesia was used, instead of allodynia in response to the test with von Frey.
· In the figures, for the Y axis, withdraw (g) has been used. Please correct it to withdrawal response, withdrawal threshold, or mechanical allodynia threshold (g). It is because withdrawal can be determined with the first aversion response to a certain amount of pressure applied, and that must be indicated as a threshold if it is the case. The authors also have thermal stimulation, and the response to that can be the thermal threshold.
· Did the authors check the normal distribution of data to use the parametric tests and for the description of data as mean and variations? Add the result of the test (authors have indicated the test in the statistics section).
· How the findings of this study can be translated for an implication in the clinic? to use the EA or drugs that can target the pathway? can it be one method superior to another or non-inferior?
